# Pulmonary Hypertension in Sickle Cell Disease: Novel Findings of Gene Polymorphisms Related to Pathophysiology

**DOI:** 10.3390/ijms25094792

**Published:** 2024-04-27

**Authors:** Sevastianos Chatzidavid, Pagona Flevari, Ioanna Tombrou, Georgios Anastasiadis, Maria Dimopoulou

**Affiliations:** Thalassemia and Sickle Cell Disease Unit, Center of Expertise in Rare Hematological Diseases (Hemoglobinopathies), Laikon General Hospital Member of EuroBlood NET, 16 Sevastoupoleos Str., 11526 Athens, Greece; sebastianx87@yahoo.com (S.C.); fpagona@yahoo.gr (P.F.); joanna.tomprou@yahoo.gr (I.T.); paokanas@gmail.com (G.A.);

**Keywords:** sickle cell disease, pulmonary hypertension, gene polymorphisms, gene sequencing

## Abstract

Pulmonary hypertension (PH) is a progressive and potentially fatal complication of sickle cell disease (SCD), affecting 6–10% of adult SCD patients. Various mechanisms and theories have been evaluated to explain the pathophysiology of this disease. However, questions remain, particularly regarding the clinical heterogeneity of the disease in terms of symptoms, complications, and survival. Beyond the classical mechanisms that have been thoroughly investigated and include hemolysis, nitric oxide availability, endothelial disorders, thrombosis, and left heart failure, attention is currently focused on the potential role of genes involved in such processes. Potential candidate genes are investigated through next-generation sequencing, with the transforming growth factor-beta (TGF-β) pathway being the initial target. This field of research may also provide novel targets for pharmacologic agents in the future, as is already the case with idiopathic PH. The collection and processing of data and samples from multiple centers can yield reliable results that will allow a better understanding of SCD-related PH as a part of the disease’s clinical spectrum. This review attempts to capture the most recent findings of studies on gene polymorphisms that have been associated with PH in SCD patients.

## 1. Introduction

Sickle cell disease (SCD) refers to a spectrum of syndromes in which hemoglobin S (HbS) is produced as a result of a point mutation in the beta-globin chain where A is replaced by T at codon 6. This mutation is inherited in a homozygous state or coinherited with a thalassemic mutation at the other beta-globin allele, leading to reduced or absent production of normal beta globin. The most common genotypes encountered include sickle cell anemia, sickle-beta thalassemia, and hemoglobin SC disease, among others.

The point mutation in the beta-globin gene, which results in SCD, leads to the production of sickle hemoglobin, which polymerizes in hypoxic conditions. The clinical manifestations of SCD cover a wide range of clinical entities. Nevertheless, the major features can be categorized based on their connection to hemolytic anemia and vaso-occlusion mechanisms. Most common complications can lead to acute and chronic illness that may have a major impact on morbidity and mortality (Figure 1).

Among pulmonary complications of SCD, pulmonary hypertension (PH) is a relatively common and usually severe condition and an independent risk factor for mortality [1,2]. The reported incidence of PH in adult SCD patients ranges between 6% and 11% [3]. The presence of PH is often suspected when exertional dyspnea is observed. Initial evaluation requires Doppler echocardiography, measurement of N-terminal pro-brain natriuretic peptide (NT-proBNP) levels, and a six-minute walk test. Definitive diagnosis is based on right heart catheterization, with demonstration of a high resting mean pulmonary arterial pressure (mPAP). Hemolytic anemia is a risk factor for the development of PH. Patients with PH have characteristic hemodynamic features, several comorbidities, and distinct phenotypes.

In an effort to address PH pathophysiology and novel findings regarding the potential influence of genetic modifications, we searched for articles indexed in PubMed using the keywords “Sickle cell disease”, “Pulmonary hypertension”, and “Genetic polymorphisms”. Through a review of the articles identified, we sampled the data for the genetic polymorphisms reported to be associated with pulmonary hypertension features in SCD patients.

## 2. Results

### 2.1. Classification of PH

According to the latest updates, patients with PH are potentially classified into one of five groups based on etiology, as presented below (Table 1) [4,5]. The first group includes patients with pulmonary arterial hypertension, while the remaining four groups include patients who are considered to have PH. Regarding SCD, some patients with consistent hemodynamic features are classified in group 5, while others demonstrate features of PH related to left-sided heart disease or thromboembolic disease and are placed in groups 2 and 4, respectively [4].

### 2.2. Clinical Presentation of PH

The diagnosis of PH in SCD patients may prove challenging in some cases. Prevalence of PH increases with increasing age, and PH is a leading cause of early death as right heart failure may have been already established when patients become symptomatic. Typical symptoms include exertional dyspnea, fatigue, chest pain, and signs of right heart failure such as lower extremity edema and palpitations [6]. On the other hand, other common accompanying conditions such as anemia, left ventricular dysfunction, and hepatic cirrhosis may be responsible for the symptoms and signs above and further complicate diagnosis.

### 2.3. Diagnostic Assessment for PH

Diagnostic steps should include factors that may also contribute to PH pathogenesis, including iron overload, chronic liver disease, HIV infection, nocturnal hypoxemia, and pulmonary thromboembolism. Initial laboratory studies usually demonstrate an elevated level of NT-proBNP [2]. Plasma NT-proBNP is a peptide released by the myocardium that can be useful in identifying patients with SCD at higher risk of PH and increased mortality risk [7,8].

Doppler echocardiography is used to screen for PH and presence of right heart failure. It is used to estimate pulmonary artery and right ventricular systolic pressures and assesses left and right ventricular size, thickness, and function. Tricuspid regurgitant velocity (TRV) measurement has been proposed as a useful tool to estimate pulmonary artery systolic pressure (PASP) [9]. Several studies have reported that using a TRV of 2.5 m/s as a cutoff point for elevated PASP practically means that 20–30% of SCD patients are diagnosed with PH, and it is suggested that even a mildly elevated TRV is associated with decreased survival [9,10,11].

A six-minute walking test with oximetry may be used to assess functional capacity and is inversely correlated with PH severity [12].

The use of pulmonary functional tests should be considered on a case-by-case basis as many SCD patients develop abnormal pulmonary function, with mild restriction patterns and abnormal diffusing capacity in the context of pulmonary fibrosis. A ventilation–perfusion scan is also necessary to exclude chronic pulmonary thromboembolic disease as it is superior to computed tomography pulmonary angiography [13]. Lastly, cardiac magnetic resonance imaging is increasingly applied and may relate right and left ventricular cardiac output and function with prognostic features in SCD patients with PH [14,15].

Right heart catheterization remains the gold standard method for definitive diagnosis and hemodynamic aspects of PH [11]. Currently, PH diagnosis is established by measuring a resting mPAP > 20 mmHg [16]. While 6 to 11 percent of SCD patients have PH based on an mPAP ≥ 25 mmHg, hemodynamic features vary across patients, with 40 percent of patients demonstrating a predominantly precapillary PH, while in the rest, the hemodynamic pattern suggests postcapillary PH [4]. Some patients have mixed hemodynamic features. Mortality is significantly higher in SCD patients with PH defined by right heart catheterization [12].

NT-proBNP can be proposed as a screening tool for PH when Doppler echocardiography is not available, as plasma NT-proBNP levels ≥ 160 pg/mL can detect PH with a sensitivity and specificity of 57 and 91 percent, respectively. Of note, in SCD cases, measurements may be confusing in patients with renal insufficiency or left heart failure [6,17,18].

In an observational study with SCD patients, patients with SCD and PH walked a shorter distance during the six-minute walking test than SCD patients without PH, and distance was inversely correlated with mPAP as measured by right heart catheterization [19]. In another study including children with SCD, during a walking test, oxygen saturation declined in 68 percent of children with an elevated TRV compared with 32 percent of those with a normal TRV [20]. However, the walking test should be cautiously used as a screening tool for PH complicating SCD because chronic anemia and other conditions may affect result interpretation.

### 2.4. Pathophysiology

Various mechanisms have been proposed to explain the occurrence of PH in SCD patients. Apart from documented findings regarding hemolysis, nitric oxide (NO) depletion, and cell-free hemoglobin, current research has now extended to the function of the activated endothelium and inflammatory response, as well as genetic polymorphisms of genes that are potentially involved in all of the above mechanisms [21,22,23,24,25,26]. Traditionally, left heart disease and diastolic dysfunction may play a key role through vascular remodeling [27] (Figure 2).

It is known that hemolysis releases high amounts of cell-free hemoglobin that overcome the binding ability of haptoglobin and hemopexin. This was initially thought to lead to massive depletion of available circulating NO, but conflicting reports on the impact of this phenomenon on NO availability exist [28]. Moreover, these effects on endothelial function through increased cell-free Hb may also provoke endothelial and vascular smooth muscle dysfunction [26].

In the case of SCD, NO bioavailability may additionally be affected by a number of other contributing factors. Increased oxidant-related metabolism of NO can reduce the vasodilating capacity of NO. NO acts mainly as a vasodilator and a platelet aggregation inhibitor. Altered redox mechanisms occur in SCD, as increased levels of oxidative molecules, including singlet oxygen, hydroxyl radical, hydrogen peroxide, and superoxide, are formed within the red blood cells of patients. These oxidative molecules can interact with NO to produce nitrate, nitrite, and peroxynitrite, which can be toxic [29].

Increased plasma levels of arginase, which is released from red blood cells, have also been proposed as a potential factor of NO availability regulation. Arginase converts L-arginine to ornithine and, therefore, limits L-arginine availability, which is an important substrate for NO production [30]. Moreover, in SCD patients, plasma arginase activity has been positively associated with secondary PH, and a lower ratio of arginine to ornithine has been associated with inflammation markers, increased soluble adhesion molecules, greater severity of PH, and an increased mortality risk [31].

The products of intravascular hemolysis, such as free heme, which is released upon hemoglobin oxidation, and microparticles containing heme species, which are collectively known as erythrocyte-danger-associated molecules (e-DAMPS), are considered to mediate oxidative stress, neutrophil extracellular trap (NET) formation, inflammasome activation, sterile inflammation, and increased endothelial adhesion.

Plasma levels of another molecule, asymmetric dimethylarginine (ADMA), may also contribute independently to endothelial dysfunction. ADMA is an NO synthase inhibitor, and in a study with SCD patients, ADMA levels correlated with hemolysis, low oxygen saturation, PH, and early death [32].

Limited NO bioavailability leads to enhanced platelet activation and endothelin-1 release. As a result, this leads to vasculopathy-related conditions that include endothelial dysfunction, vasoconstriction, inflammation, and a hypercoagulable state. These, in turn, may lead to vascular remodeling and increased pulmonary resistance with the imminent occurrence of PH features [33].

Many patients with SCD have functional asplenia or have undergone splenectomy. Splenectomy has been linked with the development of PH, particularly in patients with hemolysis. This can be partially explained by the loss of normal splenic function that also leads to platelet activation, thereby promoting pulmonary vasculature microthrombosis and endothelium dysfunction. Furthermore, after splenectomy, the rate of intravascular hemolysis increases [24].

The role of coagulation mechanisms in the pathogenesis of SCD-related PH becomes prominent. Although it is known that vascular occlusions lead to thrombosis, coagulation may be activated in patients with SCD in the absence of vascular occlusions, as suggested by increased tissue factor, increased thrombin generation, and platelet and endothelial activation. Increased circulating monocyte and endothelial cell microparticles, along with red blood cell-, platelet- and neutrophil-derived microparticles, have been reported. On the other hand, anionic phospholipids, mainly phosphatidylserine, are transported to the surface of sickle RBCs, thus enhancing coagulation activity. Soluble vascular cell adhesion molecule-1, a marker of endothelial dysfunction, is reported to correlate with the severity of hemolysis in patients with SCD-related PH [34]. The increased prevalence of thromboembolic complications has led to the acknowledgment of SCD as a prothrombotic, hypercoagulable condition [35]. In our center (Laiko General Hospital of Athens), the incidence of venous thromboembolism in SCD is as high as 23% and its presence is associated with cerebral ischemia. It is of note that D-Dimers are elevated in the vast majority of SCD patients, even in those with a steady state [36].

Obstructive sleep apnea is a frequent complication of SCD in children that is regularly underdiagnosed and has been associated with stroke and PH [37]. Existing trials in adults include a small number of patients but indicate a high incidence of sleep-disordered breathing [38]. Several mechanisms have been proposed for nocturnal oxygen desaturation in SCD [39].

Despite extensive research on the above factors’ contribution to the pathophysiology of PH in SCD patients, there are still questions that cannot be fully answered. The exact reason for which some patients exhibit severe hemolytic phenotype associated with vascular dysfunction, while in other patients, these phenomena are not so severe, cannot be fully explained, nor can patients with a high risk for vascular complications be identified easily. Advances in genome-wide investigation methods and bioinformatics appear promising for identifying gene polymorphisms associated with PH pathogenesis and prognosis.

### 2.5. Management of PH in SCD Patients

Most guidelines regarding the management and treatment of PH in patients with SCD are based on data for PH in the general population and limited expert reports on case series. It is very important to recognize and treat any SCD-related complications or comorbidities that may contribute to the clinical picture of PH. In particular, it is necessary to recognize and optimally manage the following situations:Hypoxia;Cardiopulmonary disease including left ventricular failure;Thromboembolic disorders;Nocturnal hypoxemia;Iron overload;Chronic liver disease;Chronic kidney disease.

Regular red blood cell transfusions or exchange transfusions have been reported to lower pulmonary pressures and improve functional capacity and performance status in patients with SCD and PH [40]. The correction of severe anemia is also crucial. Hydroxyurea treatment is also usually recommended to SCD patients who develop PH if they are not already on this treatment. There are no available data on the effect of other novel agents, such as voxelotor or crinzalizumab, on PH.

In clinical trials of PH-specific agents, group 5 PH patients were usually excluded, and there are insufficient prospective efficacy data on targeted markers of PH in SCD. Most initial reported data consisted of case series, and this kind of approach remains empirical and off-label use is recommended only for clinicians with expertise in the management of SCD. Sildenafil, a 5-phosphodiesterase inhibitor, has been reported to increase pulmonary pressures and function capacity in studies [41]. On the other hand, in another study, it was associated with increased painful crises and a need for hospitalization [42]. The latest data suggest that sildenafil may ameliorate symptoms based on the New York Heart Association’s functional classification, without improving the six-minute walking test or hemodynamic parameters [43]. In any case, sildenafil should be used only in patients with well-controlled disease. Riociguat, an activator of soluble guanylate cyclase, is approved for the treatment of PH, but data on its use among patients with SCD-related PH are limited [44]. Endothelin receptor antagonists, including ambrisentan and bosentan, have been evaluated in patients with SCD-related PH. They were well tolerated with improved functional capacity, mildly reduced BNP and TRV values, and, in some cases, decreased mPAP as assessed by catheterization [45]. Two randomized, double-blind studies aimed to further investigate the use of these agents. Preliminary results showed that bosentan was well tolerated, with a trend towards improved PVR [46]. Epoprostenol, a prostacyclin-targeting agent and a well-studied agent in PH, resulted in lower pulmonary pressures and PVR with improved cardiac output [47]. Moreover, prostacyclin-targeting agents may further improve functional capacity parameters [48].

Stem cell transplantation is the only curative treatment for SCD patients. A low possibility of matched sibling donor and a relative higher risk of long-term toxicities remain a concern. Gene-addition and gene-editing therapies represent a promising option and recent approvals have generated great optimism, but routine application remains limited and cost-effectiveness is not yet demonstrated. Studies are ongoing to further evaluate their effects in the future [49,50].

### 2.6. Role of Gene Polymorphisms in Pathogenesis and Prognosis of SCD-Related PH

The attempts to identify differentially expressed genes (DEGs) or single-nucleotide polymorphisms (SNPs) associated with specific complications of SCD, including PH, have been intensified with the use of next-generation sequencing techniques and genome-wide association studies. The development of PH has not been studied as extensively as other SCD complications in genomic studies, but interesting data are gradually accumulating and indicate that some of the DEGs involved are associated with mechanisms such as extracellular exosomes, platelet degranulation, blood microparticles, immune response, and protein binding. Moreover, there are indications of their involvement in crucial biological pathways such as hemopoiesis, cytokine-to-cytokine receptor interaction, TGF-β signaling pathway, and extracellular interaction mechanisms (Figure 3).

The TGF-β pathway is one of the first fields where interesting results have been reported (Ashley-Koch et al., 2008 [51]). Analyses demonstrated a potential role of genetic variation in genes related to the TGF-β pathway in PH risk. Several genes were initially implicated, including *ACVRL1*, *ADRB1*, *ADCY6*, *BMP6*, *BMPR2*, *CR1*, *FY*, *LCAT*, *LTA4H*, *SELP SERPINC1*, *SLC12A6*, and *TGFBR3*. After regression analysis, SNPs in *ACVRL1*, *BMP6*, and *ADRB1* remained significant, while *TGFBR3* demonstrated near significance. The associations with these genes are interesting because *ACVRL1* mutations have previously been reported in primary PH cases with hereditary hemorrhagic telangiectasia [52]. ACVRL1 and BMP6 belong to the TGF-β pathway. Although *BMPR2* did not remain significant in the final analysis, it has been reported elsewhere to be associated with primary PH in approximately 50% of familial PH cases [53]. In a complementary analysis in the same study, two SNPs in the ARG2 gene (rs12587111 and rs1885042) were reported to be nominally associated with PH. *ARG2* encodes arginase, which is implicated in NO dysfunction in SCD. As reported earlier, higher arginase levels are associated with PH in SCD patients [31,40].

A different study (Klings et al., 2009 [54]) reported that SNPs located in intron 1 of the NEDD4L gene may be associated with elevated serum NT-proBNP levels in SCD patients in genome-wide association studies. This gene encodes a member of the Nedd4 family of HECT domain E3 ubiquitin ligases that target specific proteins for lysosomal degradation and, therefore, is involved in the regulation of various signaling pathways, including TGF-β, autophagy, innate immunity, and DNA repair [55]. On this basis, NEDD4L may be implicated directly in SCD-related PH pathogenesis and a target for further prognostic and therapeutic application [54].

Nouraie et al. (2009) [56] investigated CYBR5 T116S polymorphism and suggested a protective effect against PH. CYBR5 T116S polymorphism was associated with lower TRVs, possibly due to the lower hemolysis levels among carriers of this SNP56. CYBR5 T116S is a frequently identified polymorphism in African children and may be associated with protection from severe malaria-related anemia. This polymorphism may be related to increased cytochrome b5 reductase activity, which, in turn, may explain its protective effect on hemolysis through increased antioxidant activity [57].

Another preliminary study (Desai et al., 2012 [58]) investigated patients with an elevated versus normal TRV and catheterization-confirmed PH and reported a significant association with five SNPs of the GALNT13 gene and a single SNP of the PRELP gene. Moreover, a trait locus upstream of the adenosine-A2B receptor (ADORA2B) gene was identified [59]. The GALNT13 protein is a member of the GalNAcT protein family, and it is suggested that it can be a potential prognostic factor in lung cancer [60]. The protein encoded by the PRELP gene is a leucine-rich repeat protein present in connective tissue extracellular matrix. The protein functions as a molecule anchoring basement membranes to the underlying connective tissue. This protein has been investigated in the pathogenesis of Hutchinson–Gilford progeria, psoriatic and rheumatoid arthritis, bladder cancer, and respiratory tract infections [61]. The ADORA2B gene encodes an adenosine receptor that is a member of the G protein-coupled receptor superfamily. ADORA2B-related signaling has been reported to be involved in solid tumor migration mechanisms as well as tumor-derived exosome-induced angiogenesis [62].

MAPK8 encodes a protein member of the mitogen-activated protein (MAP) kinase family. MAP kinases are involved in many crucial cellular processes, such as proliferation, differentiation, transcription regulation, and development. This kinase is activated by various cell signals, including response to oxidative stress, and targets specific transcription factors. The activation of this kinase by tumor necrosis factor alpha (TNF-alpha) is reported to be important for TNF-alpha-induced apoptosis. In a cohort of SCD patients, the A allele of a MAPK8 expression quantitative trait locus, rs10857560, was reported to be associated with precapillary PH. In a combined group, the homozygous state of the AA genotype of rs10857560 was characterized by lower MAPK8 expression and was present in all precapillary PH cases [63].

The glutathione S-transferase (GST) gene family encodes genes involved in certain processes, including detoxication and toxification mechanisms. GST genes are known to be upregulated in response to oxidative stress. An Egyptian study measured the frequency distribution of GSTM1, GSTT1, and GSTP1 gene polymorphisms in a group of adult SCD patients. It was demonstrated that the GSTM1 null genotype was associated with acute chest syndrome and veno-occlusive crisis, while the GSTT1 null genotype was associated with increased blood transfusion requirements. An absence of both GSTM1 and GSTT1 genes was significantly associated with pulmonary hypertension [64].

Endothelial NOS (eNOS), also known as nitric oxide synthase 3, is an enzyme encoded by the NOS3 gene located in chromosome 7. It is responsible for NO generation in the vascular endothelium; therefore, it plays a crucial role in regulating vascular tone, cellular proliferation, leukocyte adhesion, and platelet aggregation. Yousry et al. (2016) [65] genotyped a group of SCD patients for eNOS 4a/b and eNOS 786T>C polymorphisms and analyzed the results according to the severity of SCD clinical manifestations. They demonstrated that the homozygous mutant eNOS-786T>T genotype was significantly associated with a higher acute chest syndrome risk. On the other hand, the wild-type eNOS-4a/4b genotype seemed to be protective against vaso-occlusive crisis and PH. Lastly, the mutant homozygous haplotype (C-4a) was significantly associated with risks of acute chest syndrome, veno-occlusive crisis, and PH.

Thrombospondin-1 (TSP1) regulates TGF-β pathway activation and endothelial and smooth muscle cell proliferation, processes known to be affected in PAH. In familial cases of PAH, mutations involving the TSP1 gene highlighted it as a potential modifier gene in these rare cases [66]. When TSP1 SNPs were studied in SCD-related PH, univariate regression analyses revealed that rs1478604 and rs1478605 SNPs were associated with differences in pulmonary artery systolic pressure as assessed by echocardiography. These SNPs are proximal to the transcription start site; therefore, they may have a potential transcription-regulation activity on the TSP1 gene, with rs1478605 being the most possible in regulating TSP1 gene expression [67,68].

Endothelin-1 (ET-1) gene polymorphisms have been investigated extensively in the pathogenesis of various vascular diseases including SCD. The mutant T allele of the ET-1 G5665T polymorphism was reported to be associated with increased plasma ET-1 levels [69]. Khorshied et al. (2018) [70] reported that, in their study, statistical analysis comparing patients having the wild and the polymorphic genotypes demonstrated that pulmonary dysfunction in the form of pulmonary hypertension and acute chest syndrome, as well as severe vaso-occlusive crises, was more frequent in patients with the polymorphic genotypes. On the contrary, Thakur et al. (2014) [71] reported that ET-1 G5665T SNP had no significance in their study. Differences were also recorded in terms of sex-related distribution of these polymorphic genotypes.

Interleukin-1 β (IL-1 β) is a cytokine crucial for host defense responses to infection, sepsis, and injury. There are multiple studies on the role of cytokines in the pathophysiology of SCD showing increased levels of plasma IL-1β. It is believed that the release of cytokines in response to infection, endothelial cell activation, and other harmful stimuli may play a key role in the pathophysiology of SCD complications. Afifi et al. (2019) [72] reported an increased prevalence of the mutant genotype of IL-1β +3954 SNP in a group of Egyptian SCD patients. The mutant genotype was more prevalent in cases with PH. The mean ESPAP was significantly higher among patients with the TT genotype versus patients with the combined genotypes CC + CT. This study confirms the findings of Vicari et al. (2015) [73], who reported similar findings among SCD Brazilian patients.

Methyltetrahydrofolate reductase (MTHFR) is responsible for the transformation of 5,10-methylenetetrahydrofolate into 5-methyltetrahydrofolate. The MTHFR gene has 677C>T and 1298A>C functional polymorphisms, which produce a thermolabile form of MTHFR. Decreased MTHFR levels were proposed as a risk factor for deep vein thrombosis in the past. Several studies have investigated the association between MTHFR 677C>T and vascular phenomena of SCD, with conflicting results. Lakkakula et al. (2019) [74], in a meta-analysis, reported that mutant genotypes (CT + TT vs. CC) of the MTHFR 677C>T polymorphism were associated with an increased risk of vascular events in SCD patients.

RASA3 is a GTPase-activating protein that is involved in R-Ras and Rap1 activity and is related to vasculogenesis and endothelial mechanisms. When studied in SCD patients, RASA3 expression was lower in patients with PH and was associated with higher mortality. The SNP rs9525228 correlated with PH risk, higher TRV values, and higher pulmonary vascular resistance. Moreover, it was associated with precapillary PH values and decreased survival in a subgroup of patients [75]. The above results are summarized in Table 2.

## 3. Discussion

PH is one of the most severe complications of SCD. The mechanisms involved in its pathogenesis are complex and etiology is multifactorial. Established pathogenetic mechanisms include intravascular hemolysis, chronic lung involvement, hypercoagulable status, asplenia, and thromboembolic disease, but questions on how to identify patients predisposed to developing PH remain. In recent years, genome-wide association studies (GWASs) have become a new standard in genomic studies in several medical areas. The search for genetic risk factors for common diseases in an effort to explain clinical diversity has led to remarkable findings and applications in cardiovascular diseases, autoimmune diseases, and neoplasms, among others. The utilization of GWASs has been followed by deep DNA sequencing methods that play a crucial role in the investigation of genetic variants in a large number of samples from different populations, which should further extend our understanding of the differential expression of genes in normal and pathological samples.

Of course, these methods have certain limitations and problems in their application. GWASs require specifically organized and large quantities of data to avoid false-positive results. Usually, larger sample numbers are needed compared to gene-specific studies to validate data, and false-negative findings may result if SNPs are analyzed separately. Lastly, as it has been demonstrated in SCD, replication and validation of findings in groups of different ancestry and across all genotypic subgroups is important. On the other hand, for the time being, the cost of deep DNA sequencing remains high and access to these techniques and computational systems is not universal [76]. Other issues that may arise in GWASs include linkage disequilibrium with causative polymorphisms, interpretation of associations with SNPs without a known function, gene–gene interaction, gene–environment interference, and precise phenotype definition [77].

It is a reasonable approach to focus on pathways and mediators involved in the pathogenesis of SCD complications for identifying genetic modifiers. As expected, findings of genomic studies in SCD patients confirm the importance of pathways related to hemolysis, NO bioavailability, vascular remodeling, endothelial integrity, inflammation, and oxidant injury for PH pathogenesis. Most studies so far have investigated targeted candidate genes, and more comprehensive GWASs are expected to follow. To date, much emphasis has been placed on polymorphisms in genes related to the TGF-β pathway.

Another reasonable approach for future studies is to examine SCD patients for polymorphisms identified in patients with idiopathic pulmonary arterial hypertension unrelated to SCD. For example, Tang et al. (2022) [25] suggested several genes as possible markers for PH pathogenesis and potential targets for the prevention of pulmonary vascular restructure, including SLC4A1, AHSP, ALAS2, CA1, HBD, SNCA, HBM, SELENBP1, SERPINE1, ITGA2B, TEAD4, TGIF2LY, GATA5, GATA1, GATA2, and FOS. It would be interesting to investigate whether these data can be confirmed in SCD, with special attention on the ancestral composition of the patient population. The recent identification of the RASA3 gene as a candidate gene involved in the pathogenesis and prognosis of PH in SCD, as well as pulmonary arterial hypertension in non-SCD patients, is also an important finding [75]. Validation of each study with larger groups as well as broader ancestral groups is necessary to confirm causative associations.

Regarding therapeutic approaches, recent data [78] have led to the approval of sotatercept, a TGF-β signaling inhibitor for PH, by the FDA; this agent was previously shown to improve anemia and ineffective erythropoiesis in thalassemic patients [79]. These findings underscore the close interplay between hemolytic anemia and pulmonary vasculature remodeling and proliferation through TGF-β mediators and highlight the importance of identifying modifiers that may lead to targeted therapy of SCD complications in the future.

## 4. Conclusions

Ultimately, the aim of identifying genetic markers predisposing to PH may help in selecting high-risk patients for more frequent screening for PH, early introduction of disease-modifying treatments such as hydroxyurea or blood transfusions, development of new targeted therapies, and monitoring of response to treatment.

## 5. Future Directions

The future of research on SCD-related complications, including PH, must utilize every available data source from multiple centers. A typical example of an initiative in this direction was the creation of the International Hemoglobinopathy Research Network (INHERENT) in 2020, as a formally coordinated network aiming to conduct thorough investigation of genetic modifiers in hemoglobinopathies. Through a large-scale, multi-center genome-wide association study, researchers may overcome the obstacles met in previous studies related to limited sample sizes and decreased statistical power. The main targets of this effort include the identification of potential genetic modifiers; the reproduction of previous results; the adoption of universally accepted genetic, phenotypic, and clinical definitions based on validated standards; and the availability of high-quality data [80]. This kind of oriented approach to gathering, analyzing, and validating data will be of major importance. The complexity of molecular events, in addition to established pathogenetic mechanisms, must be investigated to explain the clinical diversity in SCD. Only then can research progress be focused on identifying patients at risk for specific complications and potentially changing the disease course.

## Figures and Tables

**Figure 1 ijms-25-04792-f001:**
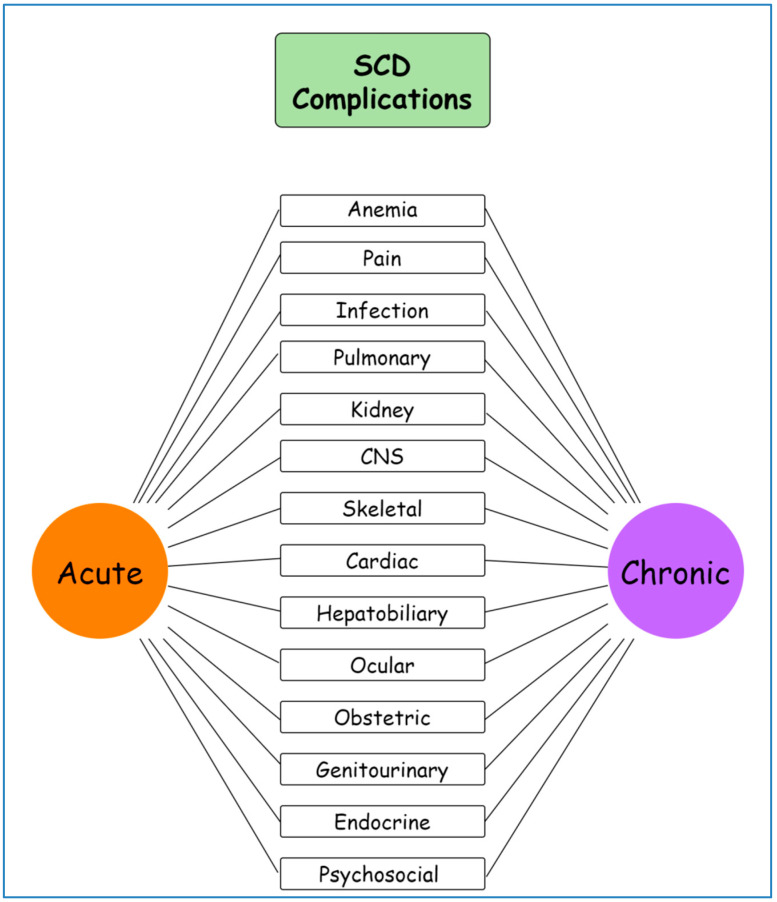
Schematic representation of sickle cell disease complications. SCD: sickle cell disease, CNS: central nervous system. Created with https://gitmind.com/app/docs, 16 April 2024.

**Figure 2 ijms-25-04792-f002:**
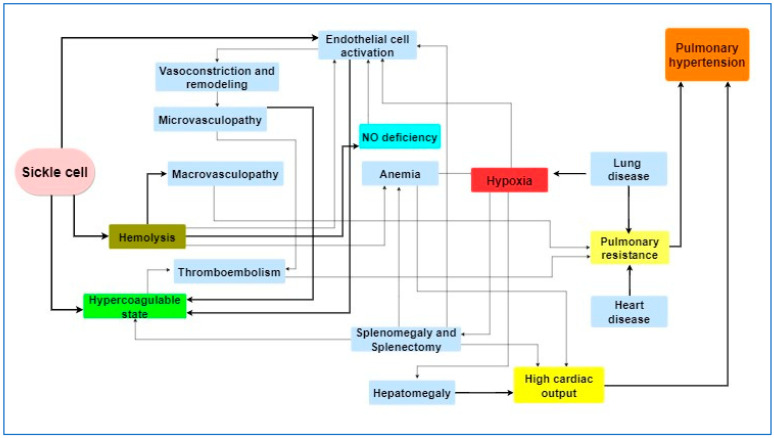
Pathophysiology of sickle cell-related pulmonary hypertension. NO: nitric oxide. (Created with https://gitmind.com/app/docs, 16 April 2024).

**Figure 3 ijms-25-04792-f003:**
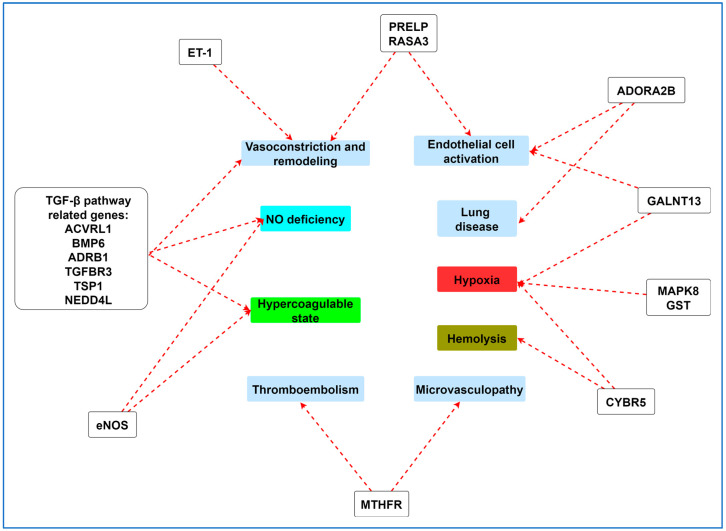
Gene polymorphisms associated with pulmonary hypertension mechanisms in SCD. TGF-β:transforming growth factor-β, ACVRL1: activin receptor-like kinase 1, BMP6: bone morphogenetic protein 6, ADRB1: beta-1 adrenergic receptor, TGFBR3: transforming growth factor beta receptor 3, TSP1: thrombospondin 1, NEDD4L: neural precursor cell-expressed developmentally downregulated gene 4-like, MTHFR: methylenetetrahydrofolate reductase, ET-1: endothelin-1, PRELP: proline/arginine-rich end leucine-rich repeat protein, RASA3: RAS P21 protein activator 3, MAPK8: mitogen-activated protein kinase 8, GST: glutathione S-transferase, ADORA2B: adenosine A2B receptor, GALNT13: polypeptide N-acetylgalactosaminyltransferase 13, eNOS: endothelial nitric oxide synthase, CYBR5: cytochrome b5 reductase, NO: nitric oxide. (Created with https://gitmind.com/app/docs, 16 April 2024).

**Table 1 ijms-25-04792-t001:** Classification of pulmonary hypertension.

**Group 1—Pulmonary Arterial Hypertension (PAH)**
1.1 Idiopathic:
1.1.1 Non-responders at vasoreactivity testing
1.1.2 Acute responders at vasoreactivity testing
1.2 Heritable
1.3 Associated with drugs and toxins
1.4 Associated with:
1.4.1 Connective tissue disease
1.4.2 HIV infection
1.4.3 Portal hypertension
1.4.4 Congenital heart disease
1.4.5 Schistosomiasis
1.5 PAH with features of venous/capillary (PVOD/PCH) involvement
1.6 Persistent PH of the newborn
**Group 2—PH Associated with Left Heart Disease**
2.1 Heart failure:
2.1.1 with preserved ejection fraction
2.1.2 with reduced or mildly reduced ejection fraction
2.2 Valvular heart disease
2.3 Congenital/acquired cardiovascular conditions leading to post-capillary PH
**Group 3—PH Associated with Lung Diseases and/or Hypoxia**
3.1 Obstructive lung disease or emphysema
3.2 Restrictive lung disease
3.3 Lung disease with mixed restrictive/obstructive pattern
3.4 Hypoventilation syndromes
3.5 Hypoxia without lung disease (e.g., high altitude)
3.6 Developmental lung disorders
**Group 4—PH Associated with Pulmonary Artery Obstructions**
4.1 Chronic thromboembolic PH
4.2 Other pulmonary artery obstructions
**Group 5—PH with Unclear and/or Multifactorial Mechanisms**
5.1 Hematological disorders
5.2 Systemic disorders
5.3 Metabolic disorders
5.4 Chronic renal failure with or without hemodialysis
5.5 Pulmonary tumor thrombotic microangiopathy
5.6 Fibrosing mediastinitis

PAH: pulmonary arterial hypertension; HIV: human immunodeficiency virus; PVOD: pulmonary veno-occlusive disease; PCH: pulmonary capillary hemangiomatosis; PH: pulmonary hypertension. European Society of Cardiology & European Respiratory Society, [5].

**Table 2 ijms-25-04792-t002:** Gene polymorphisms associated with SCD-related pulmonary hypertension.

Gene Name	SNP/Mutation	Study, Year	Number of Patients	Ancestry	PH-Related Findings
*ACVRL1*	rs3847859, rs706814	Ashley-Koch et al., 2008 [51]	518	N/A	Associated with the occurrence of PH.
*BMP6*	rs267192				
*ADRB1*	rs1801253, rs7921133				
*TGFBR3*	rs10874940				
*ARG2*	rs12587111, rs1885042				Nominally associated with PH.
*NEDD4L*	rs559046, rs1624292	Klings et al., 2009 [54]	59	N/A	Associated with a TRV ≥ 2.5 m/s.
	4 SNPs in intron 1		139		Associated with elevated NT-pro-BNP levels.
*CYBR5*	T116S	Nouraie et al., 2009 [56]	261	N/A	Heterozygosity and homozygosity for CYBR5 T116S associated with lower TRV.
*GALNT13*	rs799813, rs10497120, rs13407922, rs16833378, rs9808145	Desai et al., 2012 [58]	27	African American	Associated with elevated TRV.
*PRELP*	rs2794452				
*ADORA2B*	rs7208480				
*MAPK8*	rs10857560	Zhang et al., 2014 [63]	61	African American	Associated with precapillary PH.
*GST*	*GSTM1*, *GSTT1*, *GSTP1*	Ellithy et al., 2015 [64]	100	Egyptian	Absence of both GSTM1 and GSTT1 genes significantly associated with development of PH.
*eNOS*	eNOS 4a/b, eNOS 786T>C, C-4a	Yousry et al., 2016 [65]	100	Egyptian	Wild-type eNOS-4a/4b genotype seemed protective against VOC and PH. Mutant homozygous haplotype (C-4a) associated with the risk of ACS, VOC, and PH.
*TSP1*	rs1478604, rs1478605	Jacob et al., 2017 [67]	406	N/A	Associated with elevated TRV.
*ET-1*	G5665T	Khorshied et al., 2018 [70]	100	Egyptian	Pulmonary dysfunction (PH and ACS) more frequent in patients with the polymorphic genotypes.
*IL-1 β*	+3954	Afifi et al., 2019 [72]	50	Egyptian	Mutant genotype more prevalent in cases with PH. Mean ESPAP significantly higher among mutant genotypes.
		Vicari et al., 2015 [73]	107	Brazilian	
*MTHFR*	677C>T	Lakkakula et al., 2019 [74]	614	N/A	Mutant genotype associated with increased risk of vascular events.
*RASA3*	rs9525228	Prohaska et al., 2023 [75]	171	African American	SNP correlated with PH risk, higher TRV, and pulmonary vascular resistance, and associated with precapillary PH values and decreased survival in a subgroup of patients.

SCD: sickle cell disease, SNP: single-nucleotide polymorphism, PH: pulmonary hypertension, N/A: not available, TRV: tricuspid regurgitation velocity, ACS: acute chest syndrome, VOC: veno-occlusive crisis, ESPAP: estimated peak systolic pulmonary arterial pressure, ACVRL1: activin receptor-like kinase 1, BMP6: bone morphogenetic protein 6, ADRB1: beta-1 adrenergic receptor, TGFBR3: transforming growth factor beta receptor 3, TSP1: thrombospondin 1, NEDD4L: neural precursor cell-expressed developmentally downregulated gene 4-like, MTHFR: methylenetetrahydrofolate reductase, ET-1: endothelin-1, PRELP: proline/arginine-rich end leucine-rich repeat protein, RASA3: RAS P21 protein activator 3, MAPK8: mitogen-activated protein kinase 8, GST: glutathione S-transferase, ADORA2B: adenosine A2B receptor, GALNT13: polypeptide N-acetylgalactosaminyl transferase 13, eNOS: endothelial nitric oxide synthase, CYBR5: cytochrome b5 reductase.

## Data Availability

The data that support the findings of this study are available from the corresponding author upon reasonable request.

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
