# Peer review of "Pulmonary Hypertension in Sickle Cell Disease: Novel Findings of Gene Polymorphisms Related to Pathophysiology"

_ijms, 2024, doi:10.3390/ijms25094792_

Round 1
Reviewer 1 Report
Comments and Suggestions for Authors
- It is a well organized review and adequately informative. However, I think a small section mentioning the current management of PH in SCD should be included in the review, it can be added after the sections of diagnosis and pathophysiology. The management part is important in order to discuss the possible effects of the genetic polymorphism on the management of PH in SCD
- Figure 2 is a good summary of the topic but it is to complex for the reader given all the dashes and lines used. It is advised to make changes to the figure to simplify it to the reader to to split it into parts
Comments on the Quality of English LanguageThe English in this review is fine. It only needs minor revision
Author Response
April 23rd , 2024
Dear senior Editor and Reviewer,
We would like to submit to “International Journal of Molecular Sciences” the revised version of our review article entitled “Pulmonary Hypertension in Sickle Cell Disease: novel findings of gene polymorphisms related to pathophysiology’’.
We wish to thank you for your decision regarding our study and for allowing us to resubmit a revised version of our manuscript.
We would like to thank the reviewers for the insightful comments and suggestions, which have helped to improve our manuscript. We have carefully revised our manuscript, according to these comments. The following is a point by point response to the comments.
Reviewer’s 1 report:
- It is a well-organized review and adequately informative. However, I think a small section mentioning the current management of PH in SCD should be included in the review, it can be added after the sections of diagnosis and pathophysiology. The management part is important in order to discuss the possible effects of the genetic polymorphism on the management of PH in SCD
Response: We thank the Reviewer for these comments. Based on your comments we added a Management section after Pathophysiology section. An attempt was made to be as comprehensive as possible for reasons of space, but at the same time to provide a relatively complete update to the reader. Also a comment on the potential action of sotatercept in SCD patients based on the mechanism of action was added to the discussion.
- Figure 2 is a good summary of the topic but it is too complex for the reader given all the dashes and lines used. It is advised to make changes to the figure to simplify it to the reader to split it into parts
Response: Considering your useful comment, we proceeded to modifications regarding this point. Because the figure was loaded with a lot of information, we finally created two figures, one with the pathophysiological mechanisms and one with the connection of the genes we mention with these mechanisms.
Thank you for your consideration of this manuscript and we would like to state once again that we are eager to accept any comments and changes you would suggest after reviewing our article. The manuscript has not been previously published and has not been submitted for publication elsewhere while under consideration. All authors have read and approved the manuscript. We hope our revised manuscript will be acceptable for publication.
Sincerely yours,
Maria Dimopoulou
Thalassemia and Sickle Cell Disease Unit, Laikon General Hospital, Center of Expertise in Rare Hematological Diseases (Hemoglobinopathies), Member of EuroBlood NET
Reviewer 2 Report
Comments and Suggestions for Authors
This is a timely and reasonably comprehensive review
1. The listing of different classification groups for pulmonary hypertension in section 2.1 would be more clear if it were a distinct Table rather than sitting in the middle of the text.
2. Section 2.3, the diagnostic assessment for pulmonary hypertension, needs to be more succinct. This is important because the focus of the paper is not on the clinical diagnosis of pulmonary hypertension but rather on potential mechanisms.
3. In the discussion of gene polymorphisms and particularly in Table 1, there is some potential for confusion arising from the way in-text citations are formatted. When the authors cite a paper, they frequently put it in a parenthesis with the authors name and the study date in italics. The bracketed citation number from the reference list for that reference occurs a little farther on in the paragraph.
This leads to some confusion for the reader. I would suggest that they insert the bracketed citation number immediately behind the lead author name/date (if they need to insert the lead author name/date at all). Similarly, in Table 1, the studies cited in the table are cited by lead author name/date in italics but the bracketed reference number is not there at all. The authors should either cite both the author name/date and the bracketed reference number, or they should just cite the bracketed reference.
4. There are several places in the first few sections of the manuscript where a space between words has been omitted and the words run together. The authors should review the manuscript to correct these typographical errors.
Author Response
April 23rd , 2024
Dear senior Editor and Reviewer,
We would like to submit to “International Journal of Molecular Sciences” the revised version of our review article entitled “Pulmonary Hypertension in Sickle Cell Disease: novel findings of gene polymorphisms related to pathophysiology’’.
We wish to thank you for your decision regarding our study and for allowing us to resubmit a revised version of our manuscript.
We would like to thank the reviewers for the insightful comments and suggestions, which have helped to improve our manuscript. We have carefully revised our manuscript, according to these comments. The following is a point by point response to the comments.
Reviewer’s 2 report:
- The listing of different classification groups for pulmonary hypertension in section 2.1 would be more clear if it were a distinct Table rather than sitting in the middle of the text.
Response: Regarding this comment we made changes to the structure of this section and added a summary table with the most recent classification of pulmonary hypertension.
- Section 2.3, the diagnostic assessment for pulmonary hypertension, needs to be more succinct. This is important because the focus of the paper is not on the clinical diagnosis of pulmonary hypertension but rather on potential mechanisms.
Response: Regarding this particular comment, we have removed a paragraph and a few lines that we considered to be of secondary importance to the subject of the article.
- In the discussion of gene polymorphisms and particularly in Table 1, there is some potential for confusion arising from the way in-text citations are formatted. When the authors cite a paper, they frequently put it in a parenthesis with the authors name and the study date in italics. The bracketed citation number from the reference list for that reference occurs a little farther on in the paragraph.
This leads to some confusion for the reader. I would suggest that they insert the bracketed citation number immediately behind the lead author name/date (if they need to insert the lead author name/date at all). Similarly, in Table 1, the studies cited in the table are cited by lead author name/date in italics but the bracketed reference number is not there at all. The authors should either cite both the author name/date and the bracketed reference number, or they should just cite the bracketed reference.
Response: To avoid confusion we added bracketed citation numbers after author name/date through the body text as well as in Table 1.
- There are several places in the first few sections of the manuscript where a space between words has been omitted and the words run together. The authors should review the manuscript to correct these typographical errors.
Response: Manuscript was reviewed for these typographical errors and corrected in each case.
Thank you for your consideration of this manuscript and we would like to state once again that we are eager to accept any comments and changes you would suggest after reviewing our article. The manuscript has not been previously published and has not been submitted for publication elsewhere while under consideration. All authors have read and approved the manuscript. We hope our revised manuscript will be acceptable for publication.
Sincerely yours,
Maria Dimopoulou
Thalassemia and Sickle Cell Disease Unit, Laikon General Hospital, Center of Expertise in Rare Hematological Diseases (Hemoglobinopathies), Member of EuroBlood NET
